# Study on Establishing Degradation Model of Chip Solder Joint Material under Coupled Stress

**DOI:** 10.3390/ma13081813

**Published:** 2020-04-12

**Authors:** Longteng Li, Bo Jing, Jiaxing Hu

**Affiliations:** Prognostics and Health Management Lab, Air Force Engineering University, 1# Baling Road, Baqiao District, Xi’an 710038, China; longtengli2020@126.com (B.J.); li20201026@outlook.com (J.H.)

**Keywords:** solder joint, degradation model, failure physical model, SEM analysis, FEM simulation

## Abstract

The chip is the core component of the integrated circuit. Degradation and failure of chip solder joints can directly lead to function loss of the integrated circuit. In order to establish the degradation model of chip solder joints under coupled stress, this paper takes quad flat package (QFP) chip solder joints as the study object. First, solder joint degradation data and failure samples were obtained through fatigue tests under coupled stress. Three types of micro failure modes of solder joints were obtained by scanning electron microscope (SEM) analysis and finite element model (FEM) simulation results. Second, the characterization of degradation data was obtained by the principal component of Mahalanobis distance (PCMD) algorithm. It is found that solder joint degradation is divided into three stages: strain accumulation stage, crack propagation stage, and failure stage. Later, Coffin–Manson model and Paris model were modified based on the PCMD health index and strain simulation. The function relationship between strain accumulation time, crack propagation time, and strain was determined, respectively. Solder joint degradation models at different degradation stage were established. Finally, through strain simulation, the models can predict the strain accumulation time and failure time effectively under each failure mode, and their prediction accuracy is above 85%.

## 1. Introduction

With the development of electronic equipment reliability support, traditional time-based maintenance support technology has gradually changed to condition-based maintenance support technology. Predicting the health of electronic equipment accurately is a prerequisite for condition-based maintenance [1,2]. Whether in aviation and aerospace or other fields, as the core function component of integrated circuit, the electronic chip is the key factor to ensure the normal operation of the entire electronic equipment [3,4]. Due to the diversification of mission requirements, the service environment is more severe for electronic equipment. Chip solder joints are prone to degradation damage under coupled stress, such as vibration, shock, and temperature conversion [5,6]. Among the solder joints, a single solder joint failure can cause the entire chip to fail, and the first failed solder joint often determines the fatigue life of chip. Therefore, to establish effective degradation models of chip solder joint is of great significance for predicting the chip fatigue life, as well as the reliability evaluation and condition-based maintenance of integrated circuits.

In terms of solder joint degradation modeling, data-driven and failure physics/physics-of-failure (PoF) methods are widely used [7]. For the data-driven method, it is popular to characterize the solder joint degradation by electrical and mechanical properties. In terms of electrical properties of solder joint, Wang et al. [8,9] used field programmable gata array (FPGA) to obtain chip solder joint resistance by online measurement. Jiang Shao et al. [10] monitored the solder joint voltage signal, to determine the failure mode during the shock test. In terms of mechanical properties of solder joints, some studies use stress, strain, and micro-displacement of solder joint as detection signals. For example, Tang et al. [11] obtained the strain value of the chip solder joint in degradation by attaching a strain gauge near the key solder joint. F X Che et al. [12] used a high-speed camera to capture dynamic response of the printed circuit board (PCB) and plastic quad flat packing (PQFP) pin during the random vibration test. However, the miniaturization of the chip and solder joint have made it more difficult to detect the mechanical properties.

After degradation data measure, intelligent algorithms are used to process the degradation data, to establish the degradation model and evaluate the solder joints status. For example, Zhao Cong [13] used the two-parameter Weibull model to describe the fatigue life of the solder joints. Yiwen Long et al. [14] extracted solder joints resistivity as the characteristic parameter, and an auto regressive and moving average (ARMA) model was established as mathematical description of degradation. Lee C et al. [15] measured the radio frequency (RF) impedance of the solder joint and predicted remaining life by the support vector machine (SVM).

The data-driven degradation model has great convenience and operability because it does not need to consider the inherent degradation mechanism of the object. However, the modeling accuracy of this method is related to data acquisition and intelligent algorithms closely, and the model is based on a large amount of test data, so that it is affected by multiple outside factors, such as the test environment. Therefore, some scholars try to explain the failure behavior through the physical process of material fatigue from PoF. Some studies have observed the microstructure of solder joint at degradation and failure stage, and analyzed the crack initiation location and crack propagation mode. For example, Zhang [16] and Zhu et al. [17] researched the failure mechanism through observation of the micro crack evolution and fracture shape. Some studies used finite element simulation to analyze the origin of cracks in solder joints. For example, Huang [18] and Jiaxing et al. [19] performed QFP joint stress–strain simulations under single stress and found that the concentrated location of stress–strain was coincided with the actual crack location.

Considering solder is the adhesive, the interface regions filled by adhesive substance between the chip pin and PCB essentially affect the strength and stability of the structural elements. The degradation of the adhesive substance on such regions leads to material failure. Mainly, according to the surface damage theory of M. Frémond [20] (see also [21,22,23]), Bonetti and Bonfanti et al. [24] investigated a model for adhesive contact with friction between a thermoviscoelastic body and a rigid support. It turned out that the damage theory can be successfully used for describing adhesive contact between solids. In [25], they introduced a model describing a layered structure composed by adhesive subject to a degradation process. By an asymptotic expansion method, they derived a model of imperfect interface coupling damage and temperature evolution. Raous et al. [26,27] proposed a general framework for models, describing adhesive contact between rigid bodies. Results showed the intensity of adhesion was supposed to decrease due to progressive damage and energy dissipation. In order to study the relationship between the solder joint status and its mechanical properties, many scholars refer to the classic failure physical model to assess the solder joint health status. Some studies used fatigue models based on elastoplastic strain and energy, to assess the performance of solder joints. Athamneh and Mustafa et al. [28,29] identified the relationships between the inelastic work, plastic strain, and fatigue life by exploiting strain-based Coffin–Manson model and the energy-based Morrow model. Wu et al. [30] used the Darveaux model to predict the remaining life of solder joints. In [31], a new energy-based methodology about solder joint degradation was presented. Dissipated energy in the vicinity of newly generated fracture surfaces was calculated as the cracks propagate. Some studies have considered the coupled environment, so they have modified and improved the known failure physical models. Zhu et al. [32] modified the Anand model, which is used to predict the chip life under temperature cycling coupled with electric current. According to the failure physical model, the solder joint degradation can be studied from the physical mechanism. Chen et al. [33] introduced a coupling damage model considering both low-cycle fatigue and creep. The coupling relationship between these two failure mechanisms was investigated with the effects of creep strain rate on the ductility and the effects of damage on mechanical properties of solder joint. Generally, the results obtained by this method have the highest credibility. However, there is still much controversy about the failure mechanism of solder joints, especially under coupled stress. Because failure physical models are highly dependent on degradation mechanisms, applying the model blindly has limitations. Therefore, when studying the degradation of chip solder joints under coupled stress, a single data-driven method or single failure physics method is flawed.

In this paper, a fusion method of data-driven and failure physical model is used to analyze quantitatively the relationship between solder joint degradation and micro-strain, to establish degradation models. Degradation tests under coupled stress are performed to obtain failure samples and degradation data. Solder joint failure modes are analyzed by SEM and FEM. The PCMD algorithm is used to extract and fuse the degradation features to achieve the characterization of the solder joint degradation. Secondly, the Coffin–Manson model and Paris model are modified by the new equation parameters, which are fitted by PCMD health index and strain simulation data under coupled stress. Degradation models of the strain accumulation stage and crack propagation stage are successfully established. Strain accumulation time and failure time of solder joint are predicted. Finally, the accuracy of the models is verified based on the prediction results.

## 2. Degradation Tests under Coupled Stress

### 2.1. Test Design

The QFP solder joints of the STM32 chip are packaged on PCB, using SnPb solder, as shown in Figure 1a. The chip has 100 pins, with 0.5 mm pitch. The chip package size is 14 × 14 × 1.4 mm. The structure simulation of solder joint is shown in Figure 1b.

Capacitor are connected at the chip solder joints, and two solder joints are a group. The capacitor charging time is monitored by monitor circuit shown in Figure 2. Solder joint 1 is the tested solder joint, and solder joint 2 is the feedback solder joint. By controlling the voltage of point B in front of solder joint 1, the pin output is controlled to be high or low voltage level. When the pin outputs a high level, it is equivalent to a voltage source in front of the solder joint 1. In this way, solder joint 1 and capacitor constitute a resistor–capacitance (RC) circuit, and the capacitor is charged. Solder joint 2 is used for information feedback and determines whether data collection is finished. The charging time is affected by the resistance of the solder joint 1 and the attribute of the capacitor. When the capacitor is not changed, the charging time is only related to the impedance of the solder joint 1. Therefore, when cracks occur on solder joint structure, the charging time of the capacitor must change. By measuring the charging time, the degradation process of the solder joint can be obtained indirectly.

Electronic equipment often works in environments where random vibration loads, thermal loads, electrical stress, and other loads coexist. Vibration and temperature are the two major factors that have the great impact on the comprehensive stress experienced by electronic equipment. Structural stress is generated inside solder joint under vibration loads. The coefficient of thermal expansion (CTE) of each material of solder joint is different, so that solder joint generates thermal stress in thermal loads. The coupled stress studied in this paper is the combined effect of the above two stresses. According to GB/T 2423-2019 [34], this study conducts three groups of stress test under different coupled conditions. The test arrangements are shown in Table 1. In consideration of repeatability requirement, accelerated tests of 12 test pieces are carried out under each group, of which two test pieces are verification test pieces.

### 2.2. Micro Analysis of Failure Samples

Through SEM images, we discover that there are three failure modes (FM) in each test piece shown in Figure 3. In FM1, the crack first appears at position 1 (P1) of the solder joint and propagates toward the joint inside, along with the edge of Cu pin, causing the entire pin to fall off the solder. In FM2, cracks first appear at P1 and P2, and both propagate toward the pin’s inside. They eventually run through the entire solder joint. In FM3, the crack only appears at P3 of the Cu pin and propagates inside, causing the pin to break.

### 2.3. Finite Element Simulation

Based on the finite element software ANSYS (17.0, ANSYS Company, Pittsburgh, PA, American), the three-dimension finite element model of the test piece is established. In order to ensure the accuracy of the calculation results and the reasonableness of the calculation scale, the influences of combination reaction of solder mask, pad, and solder are ignored in modeling. At the same time, we assume that the package component is ideal [18,19,35,36]. Before the fatigue test, the residual stress of the test piece is eliminated through vibration aging method, according to GB/T 25712-2010 [37], so we do not consider the effect of residual stress during simulation. Therefore, there is no residual stress in the chip package body and no bubbles, holes, or other defects in the solder. The results of the model and mesh are shown in Figure 4.

We take the finite element simulation of test 2 as an example. All the material properties are listed in Table 2, and CTE is the abbreviation of coefficient of thermal expansion. This paper uses Anand constitutive model to describe the mechanical properties of the solder joints. There are nine material parameters in the model, as shown in Table 3.

Regarding the boundary conditions, because the test piece is fixed to the vibration table through the bolt holes located at the four corners of PCB, full displacement constraints are imposed on the inner wall of bolt holes during simulation. The direction of the vibration load is perpendicular to the PCB surface, and the power spectral density (PSD) is 0.8 g^2^/Hz, as shown in Figure 5. The temperature load is a 25 °C constant temperature field.

The results of the strain simulation of QFP solder joints on the chip under Test 2 load environment are shown in Figure 6. There is a different strain distribution of the solder joints at different positions on the chip. By comparing the strain results for each solder joint, we can find that the strain is mainly concentrated in the three locations (P1–3) where crack initiation occurs in SEM, as shown in Figure 7. Similarly, the response of the joint under random vibration in other constant temperature is analyzed, and the strain concentration area is also at the P1–3 position mostly. It shows that different coupled stresses only change the magnitude of the stress value and do not change the mode of action, the failure behavior of the solder joint does not change much.

## 3. Characterization of Solder Joint Degradation

### 3.1. PCMD Health Index

Through tests under coupled stress, capacitor charging times are obtained as the whole life degradation data of QFP. The time of collecting 10 data is defined as one period. There are 10 statistical features covering a wide range of popular time domain characteristics to be extracted from 10 data of every period, namely mean value, median value, maximum value, root-mean-square value, square root amplitude value, form factor, kurtosis factor, 5% division value, 25% division value, and 75% division value. Then, 10 sets of time domain feature values are calculated. The intrinsic energy of solder joints faults is larger than the normal one, so that the intrinsic energy features are extracted from the charging time, using the complete ensemble empirical mode decomposition (CEEMD) [38]. The obtained intrinsic mode function (IMF) data sequences are intrinsic energy features extracted from the time-frequency domain. The above time domain features and time-frequency domain features form a degradation feature matrix.

When the dimension of feature is large, some parameters overlap each other. Dynamic principal component analysis (DPCA) can fuse multidimensional features, thereby reducing the dimension of the feature parameter matrix. This method can get a more comprehensive and intuitive degradation trend [39]. The Mahalanobis distance (MD) is sensitive to slight data change and can reflect the fluctuation of parameters [40] in degradation process. Therefore, the first principal component of the feature parameter matrix is calculated by DPCA. The MD between the first principal component and the normal status data is calculated in order to analyze the similarity between the normal status and the whole degradation status. Thus, the PCMD health index is obtain by Equation (1). According to the MD, the smaller the PCMD value, the greater the similarity with the normal status. It means to be closer to the normal status, and a smaller similarity means to be closer to abnormal status.
(1)$$\mathrm{PCMD}=\sqrt{{{(D}_{i}-\mu )}^{T}S^{-1}{(D}_{i}-\mu )}$$
where D_i_ is the first principal component of feature parameter matrix; μ and S are the mean and covariance of the principal components of normal solder joint, respectively.

### 3.2. Division of Degradation Stage

FM1 is taken as an example, and the obtained PCMD health index curve is shown in Figure 8. We find that the degradation process of QFP could be divided into three states: strain accumulation stage, crack propagation stage, and failure stage. During the strain accumulation stage, strain accumulation begins to occur locally on solder joint. Since the strain does not exceed the threshold, no crack initiation occurs. Therefore, the PCMD health index remains almost unchanged at first. When the first fluctuation of the PCMD health index appears at $t_{m}$, this means indirectly that the solder joint structure has irrecoverable damage (cracks initiation). In other words, strain accumulation exceeds the threshold at $t_{m}$. During the crack propagation stage, crack spread gradually caused fluctuations in PCMD values. After slow crack growth, the conductivity of the solder joint gradually weakens. When the crack spreads to a certain extent, the PCMD value fluctuates sharply. This paper defines crack penetration time as the moment $t_{n}$. Since solder joint is in vibration environment at this time, the crack opens and closes frequently with vibration, resulting in the solder joint having weaker electrical conductivity for a period. With the abrasion of fracture being more severe, the conductivity further weakens, and the PCMD value continues to increase until the peak. Therefore, $t_{n}$ must be between the moment $n_{1}$, when the fluctuation of PCMD curve suddenly increases, and the moment $n_{1}$, when the peak value of the PCMD curve appears. However, the exact time of crack penetration is indeed difficult to measure accurately, so median value of $n_{1}$ and $n_{2}$ is defined as the approximate time $t_{n}$ of crack penetration. After $t_{n}$, the solder joint has completely broken at this stage, and the crack runs through the whole interconnection. The solder joint enters failure stage. Its reliability is basically lost, and it completely fails at a very fast rate, so we only studied the crack penetration time, $t_{n}$, instead of the peak time of PCMD.

During degradation, the differences of crack initiation positions and propagation modes can be distinguished by PCMD health index trend in Figure 9. This means that macro data of the monitoring circuit can reflect microstructure changes of the solder joints effectively. Figure 9 shows PCMD trends corresponding to three failure modes. The point **m** of each curve indicates the crack initiation, as well as the solder joint to enter crack propagation stage. Point n indicates that the solder joint is about to fail in a short time.

## 4. Degradation Modeling Based on Failure Physics

### 4.1. Modeling of Strain Accumulation Stage

At the strain accumulation stage, due to the accumulation of elastic deformation, solder joints are fatigued and cracks eventually appear. About assessment of solder joints health status, Manson’s Coffin–Manson equation based on elastoplastic strain establishes the relationship between elastoplastic strain and fatigue life of solder joints, which is widely used [41]. In this section, the Coffin–Manson equation is modified through PCMD and simulation data under coupled stress, to complete the modeling of the strain accumulation stage and predict the strain accumulation time, $t_{m}$.

From the Coffin–Manson equation, we get the following:(2)$$\frac{\Delta \varepsilon }{2}=\frac{\Delta \varepsilon_{e1}}{2}+\frac{\Delta \varepsilon_{p1}}{2}=\frac{\sigma_{f}^{\prime }}{E}{\left(2N_{f}\right)}^{b}+\varepsilon_{f}^{\prime }{\left(2N_{f}\right)}^{c}$$
where Δε is the total strain range; $\Delta \varepsilon_{e1}$ is the elastic strain range; $\Delta \varepsilon_{p1}$ is the plastic strain range; $\sigma_{f}^{\prime }$ is the fatigue strength coefficient; E is the material elastic modulus; $N_{f}$ is the fatigue life cycle number; b is the fatigue strength index; $\varepsilon_{f}^{\prime }$ is the fatigue plastic coefficient; and c is the fatigue plastic index.

Coffin–Manson has obtained fatigue strength index b=−0.12 and fatigue plasticity index c=−0.6 based on a large number of material fatigue life tests [42,43,44]. At the same time, when the solder joint is in strain accumulation stage, the elastic strain has not been transformed into plastic strain, so the plastic strain term in Equation (2) can be omitted, and Equation (2) converts into the following:(3)$$\varepsilon =\frac{3.5\sigma_{u}}{2E}N_{f}{}^{-0.12}$$
(4)$$\varepsilon =\sqrt{\frac{2}{9}[{\left(\varepsilon_{x}-\varepsilon_{y}\right)}^{2}+{\left(\varepsilon_{y}-\varepsilon_{z}\right)}^{2}+{\left(\varepsilon_{x}-\varepsilon_{z}\right)}^{2}+6(\varepsilon_{xy}{}^{2}+\varepsilon_{xz}{}^{2}+\varepsilon_{yz}{}^{2})]}$$
where ε is the equivalent strain value. The equivalent strain is the equivalent of a complex strain state to a simple unidirectional tensile or compressive state. It is formed by the proper combination of various strain components, which is equivalent to unidirectional strain, as shown in Equation (4), where $\sigma_{u}$ is the final tensile strength of the material. If the number of fatigue cycles accumulation per unit time is defined as $n_{f}$, under normal working condition, the strain value of the solder joint can be expressed as follows:(5)$$\varepsilon =\frac{3.5\sigma_{u}^{}}{2E}{\left(t_{m}\cdot n_{f}^{}\right)}^{-0.12}=\frac{3.5\sigma_{u}^{}}{2E{\left(n_{f}^{}\right)}^{0.12}}\cdot \frac{1}{t_{}^{0.12}}=A_{}\cdot \frac{1}{t_{m}^{0.12}}$$
where $t_{m}$ is the strain accumulation time of the solder joint, that is, the time corresponding to the point **m** that marks the solder joint entering the crack propagation stage; A is defined as the strain strength factor, which is determined only by the solder joint material and $n_{f}$. A has nothing to do with ε and t. The formula can be converted to the following:(6)$$t_{m}^{}=\sqrt[{0.12}]{A_{}\cdot \frac{1}{\varepsilon }}$$

According to SEM analysis, the cracks of FM1 and FM3 are only located at the SnPb solder and Cu pins, respectively. Therefore, the strain simulation and PCMD data of the solder joints with FM1 and FM3 can be used to determine the $A_{SnPb}$ and $A_{Cu}$, respectively. Firstly, the solder joints of FM1 and FM3 in 10 test pieces of every test group are classified by the PCMD trend. Then, the corresponding solder joint’s strain is extracted from the strain simulation results, and its strain accumulation time ($t_{m}$) is extracted from the PCMD health index. For multiple PCMD data curves under the same failure mode, same coupled stress, and same chip position, $t_{m}$ is the average value of each curve’s $t_{m}$. Finally, the relationship curve between ε and $1/t_{m}^{0.12}$ is fitted by the linear relationship, as shown in Figure 10, and we get the following results:(7)$$A_{SnPb}^{}=0.3637,A_{Cu}^{}=0.5339$$

For FM2 with multiple crack initiation and propagation, the first crack initiation indicates that solder joint has entered degradation stage. Due to PCMD data’s sensitivity to crack initiation, its change will be caused by first crack initiation. Therefore, $t_{m}$ of FM2 should be the strain accumulation time of first crack initiation. The relationship between the strain and the $t_{m}$ in the FM2 failure mode is as follows:(8)$$\left\{ \begin{array}{l} t_{m}^{}=\min\left\{t_{m}^{P1},t_{m}^{P2},\cdots ,t_{m}^{Pn}\right\}\\ t_{m}^{Pn}=\sqrt[{0.12}]{A_{SnPb_{}or_{}Cu}\cdot \frac{1}{\varepsilon_{Pn}}} \end{array}\right.$$
where $t_{m}^{P1},t_{m}^{P2},\cdots ,t_{m}^{Pn}$ is the strain accumulation time of cracks at various places; $A_{SnPb_{}\;{or}_{}\;Cu}$ is the material strain intensity factor of cracks; and $\varepsilon_{Pn}$ is the equivalent strain of cracks at the Nth position.

Equations (6) and (8) describe the strain accumulation stage of chip solder joints under different coupled stress and establish the relationship between crack initiation and micro strain. By obtaining the strain of the solder joint, the strain accumulation time ($t_{m}$) can be predicted.

### 4.2. Modeling of Crack Propagation Stage

The model of strain accumulation stage was completed in Section 3.1, and the prediction of strain accumulation time, $t_{m}$, was realized. If the model of crack propagation stage is established, the prediction of solder joint failure time ($t_{n}$) can be completed. From the SEM analysis, the cracks in solder joints mainly are penetrating cracks. For this type of crack propagation behavior, the Paris model with higher accuracy is selected in this paper. The basic form of the Paris model is as follows:(9)$$\frac{da}{dN}=C{\left(\Delta K_{I}\right)}^{m}$$
where da/dN is the fatigue crack growth rate; C and m are material constants. As for $\Delta K_{I}=K_{\max}-K_{\min}$, $K_{\max}$ and $K_{\min}$ are the maximum and minimum stress intensity factors in the load cycle, respectively.

This study uses a modified form of the Paris model to establish the relationship between crack propagation rate and strain amplitude. This modification takes the strain amplitude as the control parameter and is expressed as follows:(10)$$\frac{da}{dN}=C_{\varepsilon }{\left(\Delta \varepsilon \right)}^{m_{\varepsilon }}$$

In the Equation (10), $C_{\varepsilon }$ and $m_{\varepsilon }$ represent the material constant when the strain amplitude is the control parameter. Take the logarithm on both sides to obtain the following:(11)$$\ln(da/dN)=\ln C_{\varepsilon }+m_{\varepsilon }•\ln\left(\Delta \varepsilon \right)$$

The crack propagation time is calculated as follows:(12)$$t_{k}=t_{n}-t_{m}$$
where $t_{n}$ is the solder joint failure time.

If we let the crack length be L and assume that the crack spreads with uniform velocity, then we get the following:(13)$$\frac{da}{dN}=\frac{L}{t_{k}}=\frac{L}{t_{n}-t_{m}}$$

Then, Equation (11) can be rewritten as follows:(14)$$\ln(t_{k})=\ln(t_{n}-t_{m})=\ln\left(C_{\varepsilon }^{\prime }\right)+m_{\varepsilon }^{\prime }•\ln\left(\Delta \varepsilon \right)$$

In Equation (14), $C_{\varepsilon }^{\prime }=L/C_{\varepsilon }$, $m_{\varepsilon }^{\prime }=-m_{\varepsilon }$. Equation (14) shows that, in the logarithmic coordinate system, when the crack length is constant, the crack propagation time in the same material has a linear relationship with the strain amplitude. Since the parameters $C_{\varepsilon }$ and $m_{\varepsilon }$ are affected by the material, there are differences in relationship of $t_{k}$ and Δε under the materials of Cu and SnPb, respectively.

In the FM1 failure mode, cracks originate from P1 and propagate inward, which is only related to the SnPb material. After the crack propagates 0.64 mm, the solder joint is in failure, so the unidirectional crack propagation length is *L*_1_= 0.64 mm in FM1, as shown in Figure 11. At this time, the crack propagation length of the solder joint in the FM1 is constant, so $t_{k}$ and Δε have a linear relationship, and the relationship in SnPb material can be determined by using the simulation and PCMD data of solder joint whose failure mode is FM1 under coupled stress.

The solder joints of FM1 in 12 test pieces of every test group are classified by the change trend of PCMD. Then, the corresponding solder joint’s strain amplitude is extracted from the simulation results, and its strain accumulation time, $t_{k1}$, is extracted from the PCMD health index. For multiple PCMD data curves under the same failure mode, same coupled stress, and same chip position, $t_{k1}$ is the average value of each curve’s $t_{k1}$. The fitted relationship curve between and under FM1 is obtained, as shown in Figure 12:(15)$$\ln(t_{k1})=5.129-0.4469\ln\left(\Delta \varepsilon_{1}\right)$$

Then the failure time ($t_{n1}$) of the solder joint under FM1 is as follows:(16)$$t_{n1}=t_{m1}+t_{k1}=\sqrt[{0.12}]{\frac{A_{SnPb}}{\varepsilon_{1}}}+\frac{e^{5.129}}{\Delta \varepsilon_{1}{}^{0.4469}}$$

The relationship between ln(da/dN) and ln(Δε) under SnPb is as follows:(17)$$\ln(da/dN)=\ln(L/t_{k})=-12.483+0.4469\ln\left(\Delta \varepsilon \right)$$

In the FM3 failure mode, the crack runs through the entire Cu pin vertically, as shown in Figure 13, and the crack length is $L_{3}=0.12mm$. The relationship between the crack propagation time’s logarithm $\ln(t_{k3})$ and strain amplitude’s logarithm $\ln\left(\Delta \varepsilon_{3}\right)$ under FM3 is fitted by linear function, as shown in Figure 14, and we can obtain the following:(18)$$\ln(t_{k3})=3.747-0.7087\ln\left(\Delta \varepsilon_{3}\right)$$

Then the failure time $t_{n3}$ of the solder joint under FM3 is as follows:(19)$$t_{n3}=t_{m3}+t_{k3}=\sqrt[{0.12}]{\frac{A_{Cu}}{\varepsilon_{3}}}+\frac{e^{3.747}}{\Delta \varepsilon_{3}{}^{0.7087}}$$

The relationship between ln(da/dN) and ln(Δε) under Cu is as follows:(20)$$\ln(da/dN)=\ln(L/t_{k})=-12.7750+0.7087\ln\left(\Delta \varepsilon \right)$$

For FM2, in order to simplify the analysis, the interior of the Cu pin is defined as the intersection of two cracks, and both cracks terminate at this position. The crack length of solder joint under FM2 is shown in Figure 15. Then, the crack propagation time under FM2 is determined by the crack which has longer propagation time, as follows:(21)$$\left\{ \begin{array}{l} t_{k2}=\max\left\{t_{k2}^{Cu},t_{k2}^{SnPb}\right\}\\ \ln(t_{k2}^{Cu})=3.747-0.7087\ln\left(\Delta \varepsilon_{2}^{Cu}\right)\\ \ln(t_{k2}^{SnPb})=4.6590-0.4469\ln\left(\Delta \varepsilon_{2}^{SnPb}\right) \end{array}\right.$$
where, $t_{k2}^{Cu}$, $t_{k2}^{SnPb}$ are the propagation times of two cracks, respectively; and $\Delta \varepsilon_{2}^{Cu}$ and $\Delta \varepsilon_{2}^{SnPb}$ are the strain amplitudes of two cracks, respectively.

The failure time of the solder joint under FM2 is calculated as follows:(22)$$t_{n2}=t_{m2}+t_{k2}=\max\left\{\sqrt[{0.12}]{\frac{A_{Cu}}{\varepsilon_{2}^{Cu}}}+\frac{e^{3.747}}{{\left(\Delta \varepsilon_{2}^{Cu}\right)}^{0.7087}},\sqrt[{0.12}]{\frac{A_{SnPb}}{\varepsilon_{2}^{SnPb}}}+\frac{e^{4.6590}}{{\left(\Delta \varepsilon_{2}^{SnPb}\right)}^{0.4469}}\right\}$$

Equations (16), (19), and (22) model the relationship between the failure time ($t_{n}$), strain (ε), and strain amplitude (Δε), under the three failure modes of the chip solder joint. By obtaining the strain and strain amplitude of the solder joint under different coupled stress, the failure time, $t_{n}$, of the solder joint can be predicted.

### 4.3. Model Verification and Error Discussion

The test and simulation data of the other two verification test pieces in each test group are used to verify the model. Two solder joints are selected from each failure mode. After PCMD analysis on test data, the health index curve is obtained to extract $t_{m}$ and $t_{n}$. The appropriate degradation model is selected according to failure mode of solder joint. The simulation data is extracted and entered into degradation model to predict the values of $t_{m}$ and $t_{n}$. Results of test data and model output are compared and analyzed. Figure 16 shows the distribution of model prediction errors with $t_{m}$ and $t_{n}$.We can see that the prediction error rate of strain accumulation time and failure time is within 15%. In other words, its prediction accuracy is above 85%, which can meet requirements. Moreover, from Figure 16, the prediction error of $t_{n}$ is significantly higher than $t_{m}$. Considering the material of the model, the crack initiates on the surface of SnPb solder or Cu pin on crack initiation stage, without interference from other substances. However, there are intermetallic compounds (IMC) where different metal materials contact closely due to the combination reaction of the solder mask, pad, and solder. IMC can grow thicker gradually and has poor brittleness, resulting in the material resistance to failure being much lower here than in the single materials, thus damaging the solder joint life [45]. The solder joint cracks of FM1 and FM2 both pass through the IMC area during the crack propagation stage. Therefore, there is greater error for prediction of $t_{n}$. At the same time, $t_{n}$ is generated based on the median estimate of n1 and n2 in the PCMD data, so $t_{n}$ has greater uncertainty than $t_{m}$, which also input larger error to the result.

Due to poor brittleness of IMC, crack propagates more easily in IMC area, so the reality propagation speed is faster than assumed uniform speed. Obviously, this means the solder joints of FM1 and FM2 reach $t_{n}$ faster in the reality situation. In order to study the IMC impact, the test and model value of $t_{n}$ is extracted for FM1 and FM2; it can be seen that test values of $t_{n}$ are indeed slightly smaller than model output values in most cases, as shown in the Figure 17. FM1 solder joint crack pass through the longest distance in IMC during crack propagation, according to SEM. FM3 solder joint crack appears only on Cu pin without the impact of IMC. Therefore, the IMC impact of on model error of FM1, FM2, and FM3 is different. The error rate average of FM1, FM2, and FM3 are calculated, and the result shows that the numerical relationship is FM1 > FM2 > FM3, as shown in Figure 18. In summary, for the degradation model of the FM1 and FM2 solder joints, the IMC is indeed an important source of model errors. Moreover, for the solder joint of FM1, the model error from IMC is largest.

## 5. Conclusions

A fusion method of data-driven and physical model is used to conduct QFP solder joint degradation model and life prediction research. Through the degradation test under the coupled stress, degradation data and failure samples are obtained. Through SEM analysis and FEM analysis, it is concluded that chip solder joints have three failure modes. Then, the PCMD algorithm is used to feature fusion and degradation characterization. The PCMD health index trends under different failure modes are analyzed. Through the strain accumulation time and crack propagation time, the degradation process under each failure mode can be divided into three stages: strain accumulation stage, crack propagation stage, and failure stage. The Coffin–Manson model and Paris model are modified by PCMD and simulation data, to establish degradation models at strain accumulation stage and crack propagation stage under each failure mode. In this way, a suitable degradation model can be chosen based on the failure mode of solder joint through finite element simulation before the chip works. Using the strain simulation data as model input, the crack initiation time and failure time of each chip solder joint can be predicted. Through verification experiments, the error rate of the model is about 15%, and the main source of errors is the IMC impact not being considered in the modeling process.

## Figures and Tables

**Figure 1 materials-13-01813-f001:**
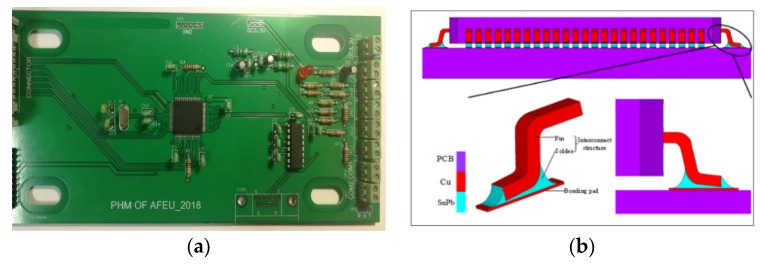
(**a**) Test piece; (**b**) quad flat package (QFP) solder joint.

**Figure 2 materials-13-01813-f002:**
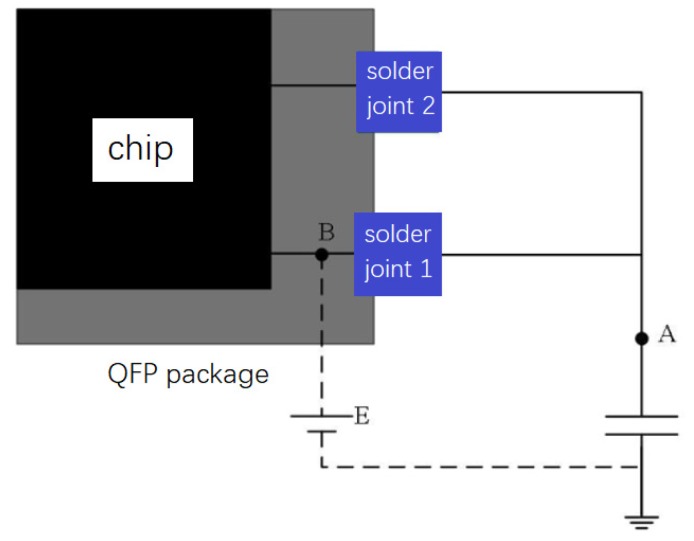
QFP solder joint detection circuit.

**Figure 3 materials-13-01813-f003:**
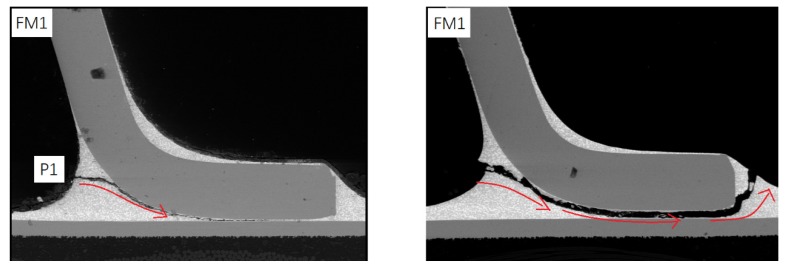
Microscopic failure modes of solder joints.

**Figure 4 materials-13-01813-f004:**
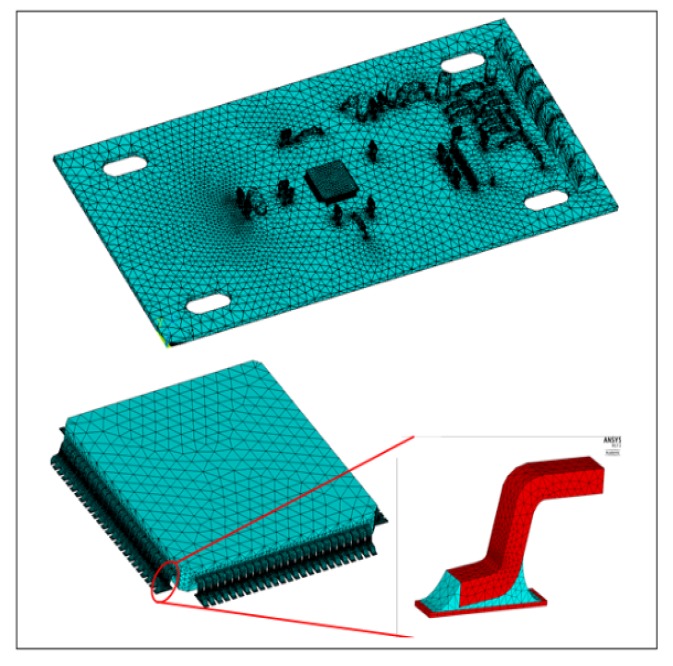
Finite element model and mesh.

**Figure 5 materials-13-01813-f005:**
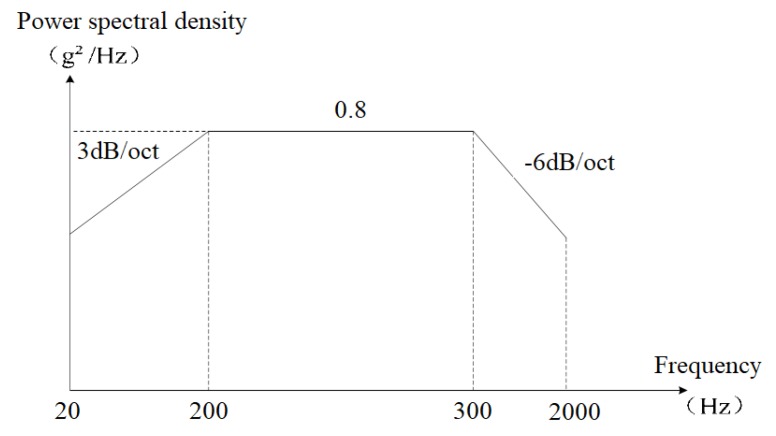
Power spectral density.

**Figure 6 materials-13-01813-f006:**
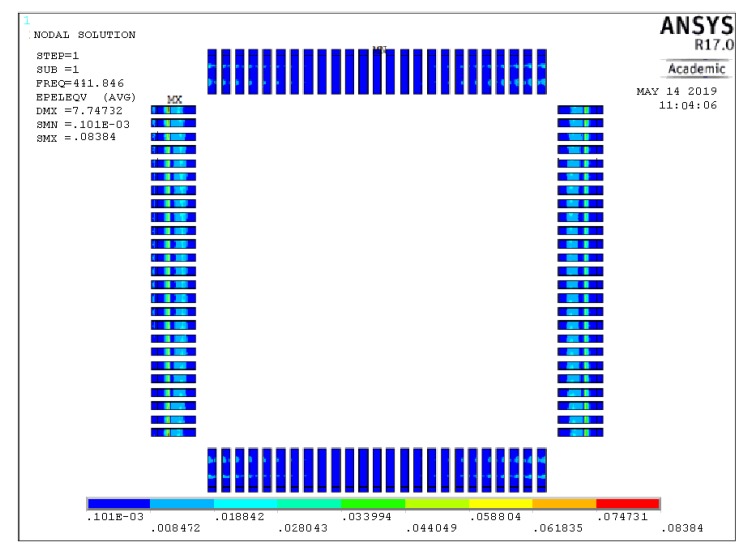
Strain distribution of all chip solder joints.

**Figure 7 materials-13-01813-f007:**
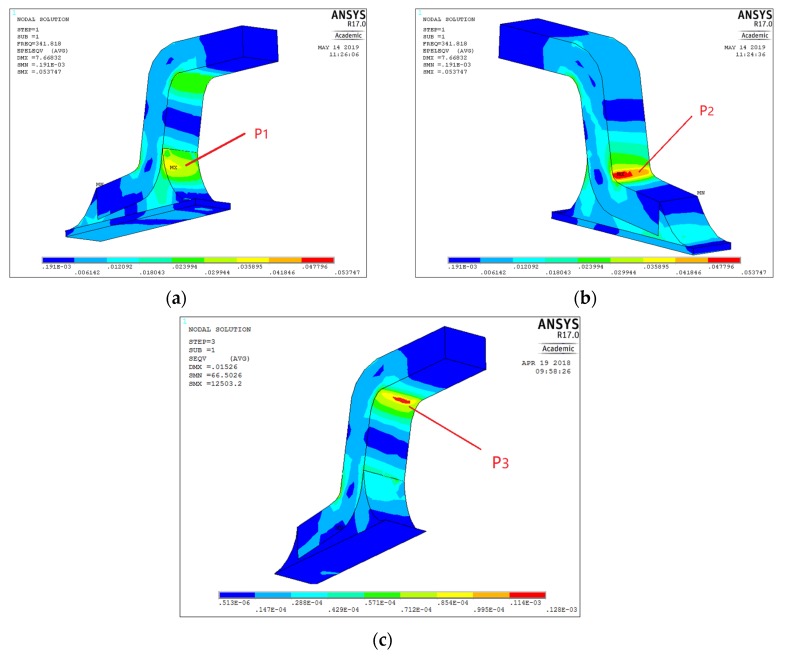
Strain simulation of single solder joint. (**a**) Strain concentration in location P1 (**b**) Strain concentration in location P2 (**c**) Strain concentration in location P3.

**Figure 8 materials-13-01813-f008:**
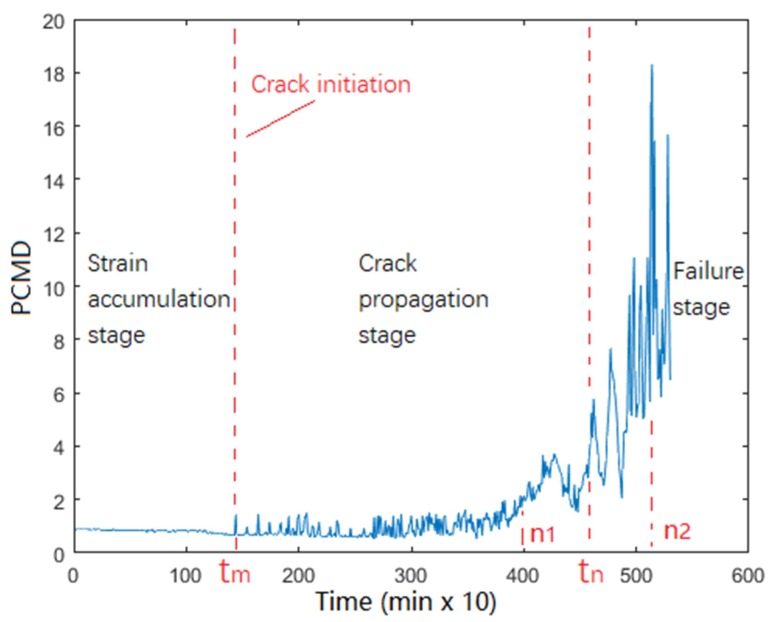
Principal component of Mahalanobis distance (PCMD) health index curve of FM1.

**Figure 9 materials-13-01813-f009:**
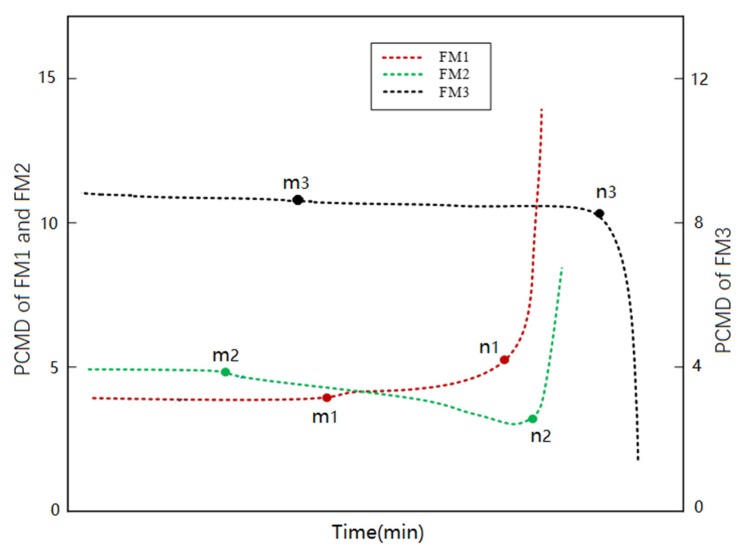
PCMD trend of each failure mode.

**Figure 10 materials-13-01813-f010:**
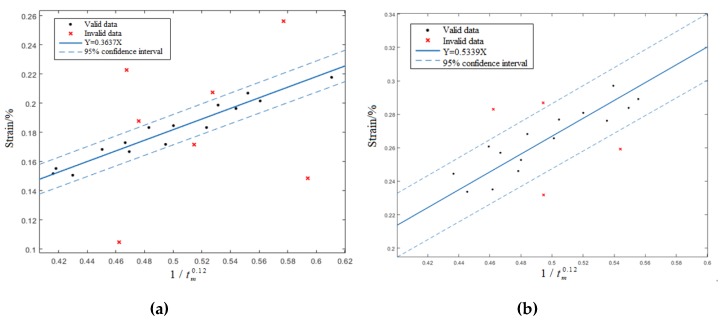
(**a**) Fitting diagram of ε and $1/t_{m}^{0.12}$ under FM1; and (**b**) fitting diagram of ε and $1/t_{m}^{0.12}$ under FM3.

**Figure 11 materials-13-01813-f011:**
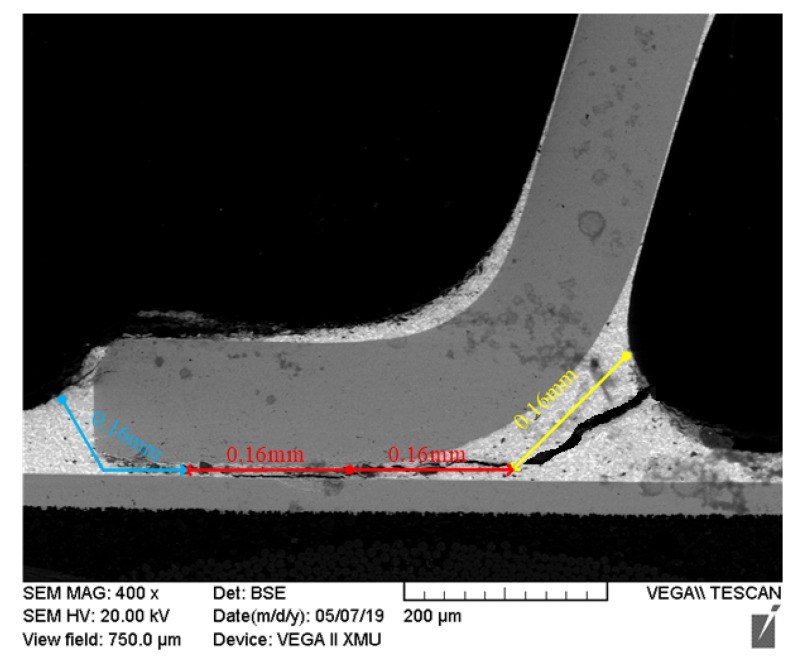
Crack length of solder joints under FM1.

**Figure 12 materials-13-01813-f012:**
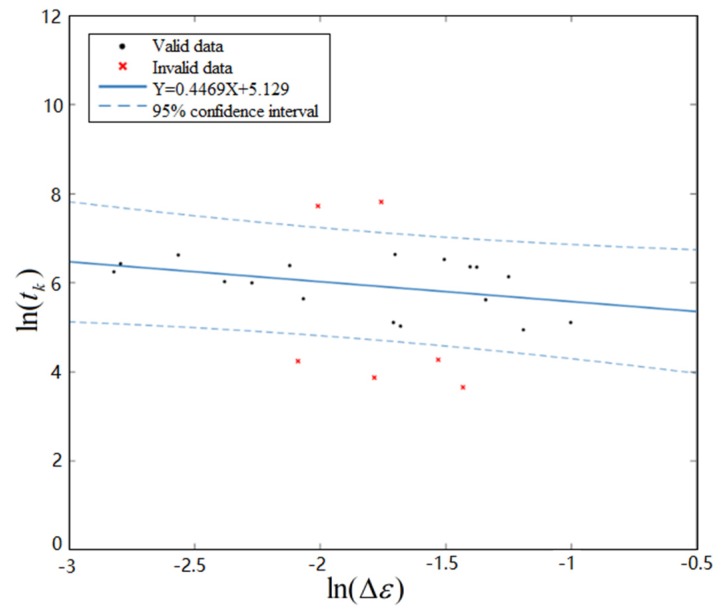
Fit curve of $\ln(t_{k})$ and ln(Δε) under FM1.

**Figure 13 materials-13-01813-f013:**
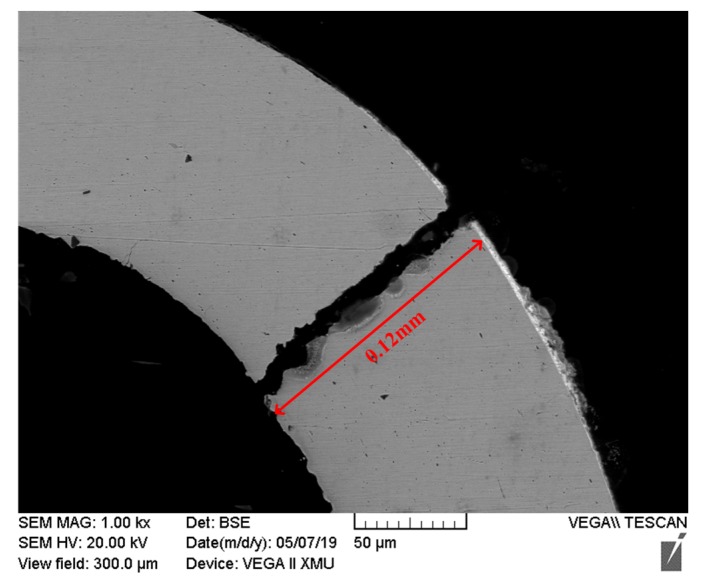
Crack length of solder joints under FM3.

**Figure 14 materials-13-01813-f014:**
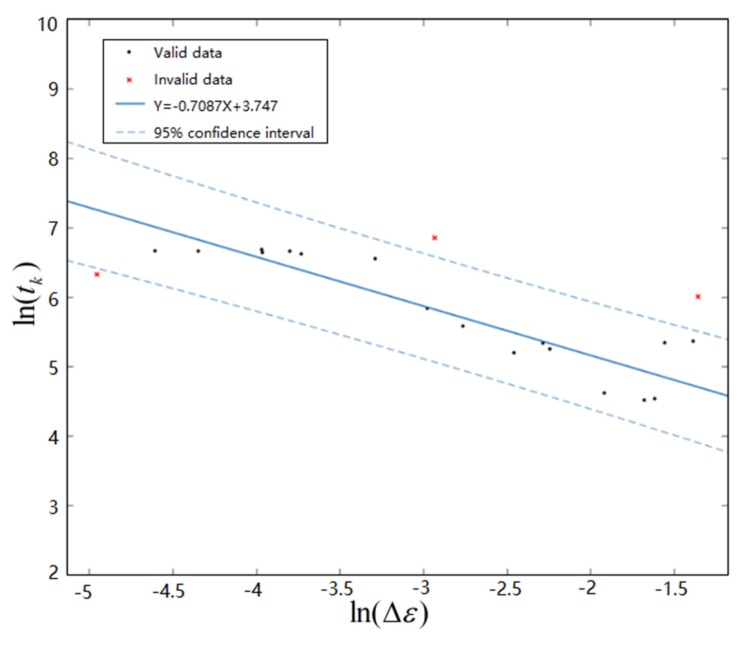
Fitting curve of $\ln(t_{k})$ and ln(Δε) under FM3.

**Figure 15 materials-13-01813-f015:**
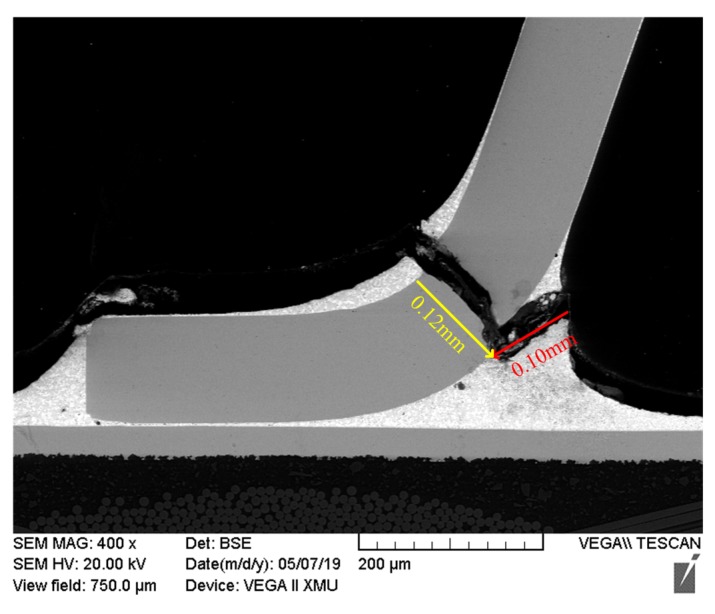
Crack length of solder joints under FM2.

**Figure 16 materials-13-01813-f016:**
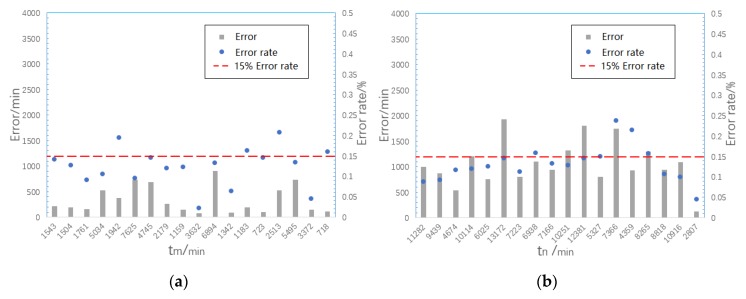
(**a**) Model prediction error distribution of $t_{m}$ (**b**) Model prediction error distribution of $t_{n}$.

**Figure 17 materials-13-01813-f017:**
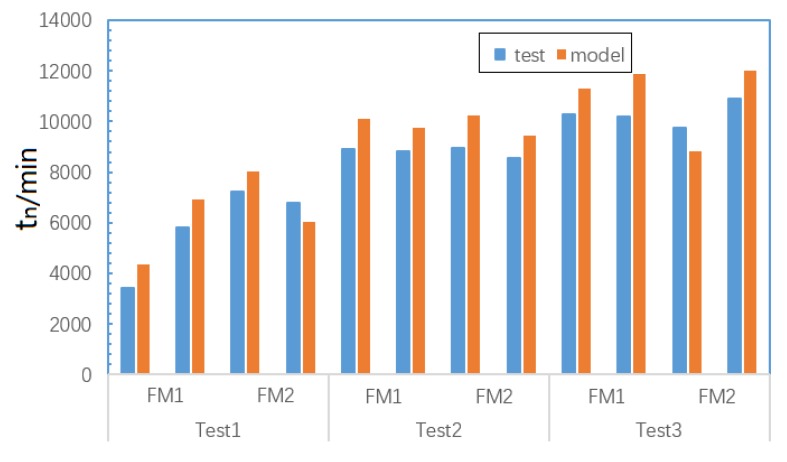
Test and model value of $t_{n}$ for FM1 and FM2.

**Figure 18 materials-13-01813-f018:**
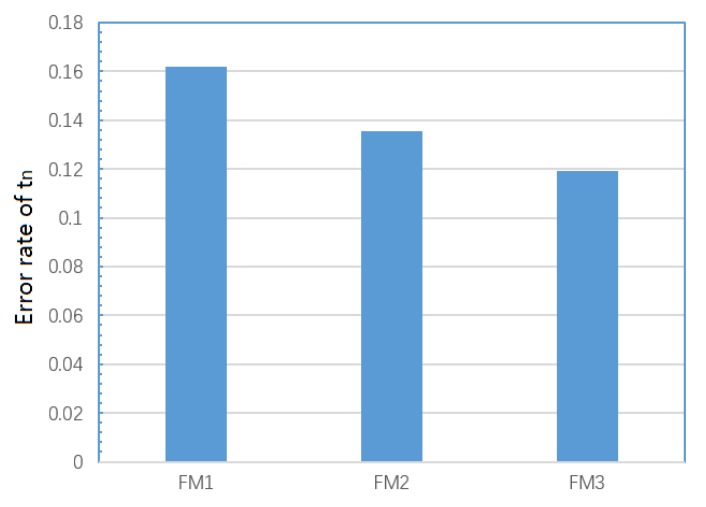
Error rate average of FM1, FM2, and FM3.

**Table 1 materials-13-01813-t001:** Degradation test scheme under coupled stress.

Test Number	Temperature Load (°C)	Random Vibration Load (g^2^/Hz)
1	−25	0.8
2	25	0.8
3	75	0.8

**Table 2 materials-13-01813-t002:** Material properties.

Material	Young’s Modulus (GPa)	Poisson’s Ratio	Density (kg/m^3^)	CET (1/K)
PCB(Fr4)	22	0.40	1900	21 × 10^−6^
Pin (Copper)	141	0.35	8700	16.6 × 10^−6^
Chip (Silicon)	17	0.28	2329	2.6 × 10^−6^
Solder (60Sn-40Pb)	30	0.40	9000	21 × 10^−6^

**Table 3 materials-13-01813-t003:** Material parameter of Anand model.

Material Parameter	Value	Definition
A/s^−1^	6220	Pre-exponential Factor
Q/(J·mol^-1^)	6310	Activation Energy/Boltzmann’s Constant
ξ	3	Multiplier of Stress
m	0.27	Hardening Constant
h_0_/MPa	60599	Strain Rate Sensitivity of Stress
a	1.8	Strain Rate Sensitivity of Hardening
ŝ/MPa	36.86	Coefficient for Deformation Resistance Saturation Value
n	0.022	Deformation Resistance Value
$s_{0}$/MPa	3.15	Initial Value of Deformation Resistance

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
