# Peer review of "Study on Establishing Degradation Model of Chip Solder Joint Material under Coupled Stress"

_materials, 2020, doi:10.3390/ma13081813_

Round 1

Reviewer 1 Report

This is very important from the practical point of view of the problems of Establishing Degradation Model of Chip Solder Joint Material Under Coupled Stress. The paper is original and present interesting information. Strong side of this paper is a research character. 

I believe that the paper can be published in Materials if the authors improve the discussion on the results. The discussion of the results is poor. The authors try to describe what it can be seen on the graphs without any real explanation of the results. They make assumptions that are not backed-up by any references, provide the results without any theory and as a result, the conclusions are rather trivial. Additionally currently, the discussions seem to be "punctual", i.e. limited only to the current results, but these need to be also open up as comments of wider academic meaning. Please try to address this. 

Reviewer 2 Report

The paper deals with a degradation model of chip solder joint material.

The paper is interesting and well written.

The results look new.

There is, however, one important point that needs to be improved. The bibliography mentions a methodology to model joint degradation. This bibliography seems very poor to me and ignores for example all the models based on the concept of "adhesion intensity" which are often very relevant,

see works of Fremond, Bonetti, Raous, Lebon, Freddi, Bonfanti, Sacco, ...

Note : equivalent strain has to be defined precisely

Author Response

Response to Reviewer 2 Comments

Dear Editor and Reviewers,

We greatly appreciate the comments and detailed reviews for our manuscript. These comments are valuable and helpful for us to improve the paper. We have carefully revised the manuscript, and also included a detailed list of revisions along with our response to the reviewer comments as follows. All changes have been marked in blue in the revised manuscript. Please find our responses to each of the comments as below:

General: The paper deals with a degradation model of chip solder joint material. The paper is interesting and well written. The results look new.

Response: Thanks so much for your deeply interests on our paper.

Point 1: The bibliography mentions a methodology to model joint degradation. This bibliography seems very poor to me and ignores for example all the models based on the concept of "adhesion intensity" which are often very relevant, see works of Fremond, Bonetti, Raous, Lebon, Freddi, Bonfanti, Sacco, ...

Response: Thanks a lot for your good suggestion. Some references have been adjusted and added in part of model joint degradation especially models based on the concept of "adhesion intensity".

Point 2: Equivalent strain has to be defined precisely

Response: Thanks so much for your great reminder. Equivalent strain has been defined precisely by text description and formula in the article. Text description has been rewritten as “Equivalent strain is the equivalent of a complex strain state to a simple unidirectional tensile or compressive state. It is formed by proper combination of various strain components, which is equivalent to unidirectional strain.”

Finally, we would like to thank again both editor and the anonymous reviewers for their time and efforts in reviewing paper. Please feel free to contact us if you have any additional questions.

Sincerely,

longteng li

Reviewer 3 Report

2020-03-26 materials
(1) Analysis of soldering failure and degradation is one of the biggest topics of the semiconductor industry, and many papers regarding this topic can be found. Because of that, I would like to ask the authors to clarify their originarity and contribution in a better way so that what is the unique contribution of the paper and how the discussed knowledge in the paper agrees or disagrees with existing ones.

(2) Meaning of "coupled stress" seems to be unclear. What is the fundamental factors dominating that femenon? For example, if the stress occurs due to the temperature change and difference of material properties, those factors should be explained clearly.

(3) The physical validity of the tested condition shown in table 1 and 2 is not clear. In the tables, the temperature load, vigration amplitude, electrical stress and material properties are plotted, but length of the transient time is not mentioned. Was that a statistic test in which the state of the test peice has been changed from an initial condition to each of the explained conditions in table 1? Were there any reasons for chosing the three test conditions in table 1? Since the obtained data is used for statistic analysis in the following section 3, the condition should be enough valid so that it covers the practical situation, but such consideration has not been discussed in the paper.

(4) In addition to the above question (3), in the presetnt paper, how the FEM simulation has been carried out is not explained except the graphics of the meshed model. How the coupled stress condition in table 1 has been reproduced?

(5) Even though the tested condition seems to be static, as I mentioned in the above question (3), the obtained results plotted in figure 6 shows a vibration behavior. In deed, the authors said "When the crack spreads to a certain extent,
the PCMD value fluctuates sharply." The source of this dynamic behavior should be explained clearly together with more clear explanation of how the test condition has given.

(6) From the explanation of figure 6, how the authors have judged the moment of t_m and t_n where the status of the failure stage changes among the three states
. This is because there is no explanation of the threshold regarding the plot to determine those moments.

Reviewer 4 Report

The paper deals with the performance degradation of chips due to solder joint erosion; the degradation is analyzed indirectly by a data-driven/mechanical failure model. The paper is well structured and of significance; the methodology is well explained. However, there are many assumptions that need further explanation, especially the fatigue fracture part.

  • line 40: What is FPGA? Also, PQFP, ARMA… There are numerous abbreviations, so please be sure to explain them upon first occurrence.
  • 129: The influences of the solder resistance layer, the combination reaction of pad and solder are ignored in modeling. At the same time, we assume that the package component is ideal.
    Please discuss the (potential) ramifications of these assumptions on the results.
  • FEA/Table 2 gives only elastic properties. Is this a purely elastic analysis? Further, residual stresses may have an enormous impact on the stress distribution. Please provide boundary and loading conditions.
  • 138-140: What do you mean by “strain distribution positions?
  • 5: What are the differences between the 3 screenshots? Location, loading, temperature, Table1…?
    Please consider rewriting the paragraph 137-143 and give more details of the FE modelling.
  • 180, Fig.6/7: “During the crack propagation stage, the strain accumulate exceeds the threshold at tm.” It is not clear how this threshold is defined. The PCMD curve remains flat up to much higher t. The same applies to tn.
  • 220: Are these literature values? If so, please cite. Else, provide details.
  • 9: Replacing the SIF amplitude by the strain amplitude neglects the specific concept of the SIF, which represents the stress distribution around the crack tip. Further, which strain amplitude is this, what does it represent? It can hardly be the very crack tip.
  • 282: How do you justify the assumption of constant crack propagation speed?
  • 9 suggests interlaminar failure; the crack propagates along the border (red line). The material resistance to failure will be much lower here than in the single materials. Please discuss this. The adhesion layer may have to be modelled separately.
  • The model validation part is too short and there is no discussion part. The many assumptions mean some uncertainty; error sources and their impacts should be discussed.
  • English language is ok.

Round 2

Reviewer 2 Report

The paper is now acceptable for publication

Reviewer 3 Report

After the revision, the paper seems to be written carefully and enough informative. The authors has paid great effort for answering my questions and revising the paper. I thus do not have any further question or comment at this time.